# miR-27a-5p Attenuates Hypoxia-induced Rat Cardiomyocyte Injury by Inhibiting *Atg7*

**DOI:** 10.3390/ijms20102418

**Published:** 2019-05-16

**Authors:** Jinwei Zhang, Wanling Qiu, Jideng Ma, Yujie Wang, Zihui Hu, Keren Long, Xun Wang, Long Jin, Qianzi Tang, Guoqing Tang, Li Zhu, Xuewei Li, Surong Shuai, Mingzhou Li

**Affiliations:** 1Institute of Animal Genetics and Breeding, College of Animal Science and Technology, Sichuan Agricultural University, Chengdu 611130, Sichuan, China; Jinweizhang50@163.com (J.Z.); qiuwanling2016@163.com (W.Q.); jideng.ma@sicau.edu.cn (J.M.); wangyujie715@163.com (Y.W.); Huzihui2016@163.com (Z.H.); keren.long@sicau.edu.cn (K.L.); xun_wang007@163.com (X.W.); longjin8806@163.com (L.J.); wupie@163.com (Q.T.); tyq003@163.com (G.T.); zhuli7508@163.com (L.Z.); xuewei.li@sicau.edu.cn (X.L.); 2Farm Animal Genetic Resource Exploration and Innovation Key Laboratory of Sichuan Province, Sichuan Agricultural University, Chengdu 611130, Sichuan, China

**Keywords:** miR-27a-5p, acute myocardial infarction, autophagy, apoptosis, hypoxia

## Abstract

Acute myocardial infarction (AMI) is an ischemic heart disease with high mortality worldwide. AMI triggers a hypoxic microenvironment and induces extensive myocardial injury, including autophagy and apoptosis. MiRNAs, which are a class of posttranscriptional regulators, have been shown to be involved in the development of ischemic heart diseases. We have previously reported that hypoxia significantly alters the miRNA transcriptome in rat cardiomyoblast cells (H9c2), including miR-27a-5p. In the present study, we further investigated the potential function of miR-27a-5p in the cardiomyocyte response to hypoxia, and showed that miR-27a-5p expression was downregulated in the H9c2 cells at different hypoxia-exposed timepoints and the myocardium of a rat AMI model. Follow-up experiments revealed that miR-27a-5p attenuated hypoxia-induced cardiomyocyte injury by regulating autophagy and apoptosis via *Atg7*, which partly elucidated the anti-hypoxic injury effects of miR-27a-5p. Taken together, this study shows that miR-27a-5p has a cardioprotective effect on hypoxia-induced H9c2 cell injury, suggesting it may be a novel target for the treatment of hypoxia-related heart diseases.

## 1. Introduction

Acute myocardial infarction (AMI) is often the primary pathological cause of death and disability worldwide [1]. During AMI, acute occlusion of the coronary artery deprives the oxygen and nutrients in myocardium and will contributes to cardiac dysfunction, including hypertrophy and remodeling, eventually leads to heart failure [2]. Since cardiomyocytes are terminally differentiated cells that have no or little regenerative potentialities, thus preventing cardiomyocytes loss after AMI injury is clinically a vital therapeutic strategy. Cardiomyocytes death and survival are affected predominantly via three cellular pathways: apoptosis, necrosis and autophagy [3]. Out of these three cellular pathways, apoptosis and necrosis have been extensively researched in AMI, but the effect of autophagy underlying AMI is still controversial to date [4]. Autophagy is an evolutionarily conserved process that maintains homeostasis in a cellular response to stresses by degrading abnormal protein and damaged organelles, which is considered to be closely associated with many heart diseases such as AMI [5]. Recently, autophagy has been considered a double-edged sword in the context of AMI, i.e., autophagy in early stage of AMI is beneficial to cardiomyocytes survival but excessive autophagy after AMI will induce autophagic cell death [6]. Thus, it is indispensable to further elucidate the autophagy regulation mechanism in cardiomyocytes survival after AMI.

MicroRNAs (miRNAs), a class of highly conserved non-coding RNAs, are major posttranscriptional regulators that involving in almost all cellular processes [7]. Currently, accumulating evidence has shown that miRNAs play essential roles in some heart diseases by regulating autophagy-related genes [4]. miRNA-212/132 family induce both cardiac hypertrophy and heart failure by activating pro-hypertrophic calcineurin/NFAT signaling, while inhibiting autophagic response upon starvation by directly targeting the anti-hypertrophic and pro-autophagic FoxO3 transcription factor [8]. miR-188-3p inhibits autophagy and autophagic cell death in the heart by targeting *Atg7* expression, meanwhile this effect can be suppressed by lncRNA APF (autophagy promoting factor) [9]. miR-21 alleviates hypoxia/reoxygenation-induced injury in H9c2 cells through weakening excessive autophagy and apoptosis via the Akt/mTOR pathway [10]. miR-204 has a protective effect against H9c2 cells hypoxia/reoxygenation-induced injury by regulating SIRT1-mediated autophagy [11]. Moreover, a recent study reports that miR-223 alleviates hypoxia-induced excessive autophagy and apoptosis in rat cardiomyocytes via the Akt/mTOR pathway by targeting *PARP-1* [12]. These miRNA may be serve as a potential target for ischemic heart disease treatment. In our previous study, we noted that the expression of miR-27a-5p decreased in acute hypoxia-exposed H9c2 cells using a small RNA-seq [13]. However, whether miR-27a-5p affects hypoxia-induced cardiomyocyte survival through regulating cell autophagy after AMI are still unknown.

In this study, we established a model of hypoxia in H9c2 cells and developed an AMI model in the rat to investigate the miR-27a-5p expression pattern in H9c2 cells and the main visceral tissues of rats (Figure 1). We found that hypoxia induced cell injury in vivo and in vitro and was accompanied by downregulation of miR-27a-5p expression. miR-27a-5p upregulation attenuated hypoxia-induced cardiomyocyte injury by regulating autophagy and apoptosis via *Atg7*, suggesting that miR-27a-5p may be a novel treatment strategy for hypoxia-related heart diseases.

## 2. Results

### 2.1. Hypoxia Induces H9c2 Cells Injury and Reduces miR-27a-5p Expression

In this study, we first cultured H9c2 cells in hypoxic condition for 24 h to simulate hypoxia induced by AMI *in vitro*. We found that hypoxia increased HIF-1α protein expression (Figure 2A) and triggered cell injury, including a decrease in cell viability (*p* < 0.01; Figure 2B), increased cell membrane damage (*p* < 0.01; Figure 2C) and apoptosis and necrosis (*p* < 0.01; Figure 2D,E). Meanwhile, hypoxia significantly increased the expression of proapoptotic genes (*Caspase-3*, *BAX*, *Faslg* and *P53*, *p* < 0.01; Figure 2F), but decreased expression of the antiapoptotic gene *Bcl-2* (*p* < 0.05; Figure 2F). Autophagy has previously been observed in ischemic heart disease [14,15] and autophagy levels were assessed in hypoxia-exposed H9c2 cells by western blot and autophagosome formation. These data showed that hypoxia increased autophagosome formation (Figure 2G) and promoted the switch of LC3-I to LC3-II. It also resulted in a reduction in P62 protein expression (*p* < 0.01; Figure 2H). Next, miR-27a-5p expression pattern was assessed in hypoxia-exposed H9c2 cells using qRT-PCR. miR-27a-5p expression decreased in a time-dependent manner (Figure 2I). These results indicate that hypoxia induced cell injury and reduced miR-27a-5p expression levels in H9c2 cells.

### 2.2. AMI Triggers Widespread Injury Accompanied by Downregulation of miR-27a-5p in Rats

To investigate whether the miR-27a-5p expression under hypoxia induced by AMI in vivo was similar to that in hypoxia-exposed cardiomyocytes in vitro, an AMI rat model was established by ligating the coronary artery [16]. We observed S-T segment elevation in the electrocardiogram (ECG) and a reduction in blood pressure (BP) in the AMI group compared with sham, which confirmed successful AMI (Figure 3A,B). A *post hoc* power analysis of ∆ BP obtained a power of > 0.90 with *p* = 0.05 in every LAD ligation timepoint (see “Statistical Analysis” for details on power analysis) (Appendix A). We also found that the organ index in several main visceral tissues (except lung) was reduced (Appendix A), which may be associated with the decreased left ventricular ejection fraction commonly observed after AMI [12]. Meanwhile, HE staining of the left ventricle showed that the cells in sham rat hearts were arranged uniformly with a normal gap, but local necrosis (indicated by arrowhead) and intercellular gaps (indicated by asterisk) were observed in AMI rats (Figure 3C). These data indicate that AMI induced severe damage in the rat myocardium.

The expression pattern of apoptosis-related genes showed that AMI triggers widespread apoptosis in the main visceral tissues, especially heart (*p* < 0.01), compared with sham (Figure 3D & Appendix A). AMI also increased HIF-1α expression, shifted the expression of LC3-I to LC3-II, and decreased the expression of P62 protein (Figure 3E), which indicates that AMI synchronously promotes autophagy and apoptosis in the rat myocardium. In addition, AMI caused a reduction in miR-27a-5p expression in several visceral tissues, in particular the heart and kidney (*p* < 0.01; Figure 3F) when assessed by qRT-PCR analysis. The above results indicate that, similar to the in vitro results, hypoxic injury is widely induced in AMI rats and is accompanied by widespread downregulation of miR-27a-5p. Thus, miR-27a-5p may play a role in AMI-induced hypoxic injury.

### 2.3. Upregulation of miR-27a-5p Attenuates Hypoxia-Induced Excessive Autophagy and Apoptosis

Several studies have previously reported that autophagy and apoptosis successively appear in the cardiovascular diseases and the crosstalk between them plays an important role in the development of ischemic heart disease [17,18]. Autophagy have bidirectional effects in AMI, as autophagy may have both damaging and protective roles depending on the hypoxic conditions, such as duration or severity [19]. In the present study, assessment of autophagic flux showed that hypoxia-exposed H9c2 cells increased the level of autophagy in a time-dependent manner (Appendix A). Next, cell viability and membrane damage were assessed after hypoxia in H9c2 cells pretreated with 10 mM 3-MA (a widely-used autophagy inhibitor). Cell viability was decreased in 3-MA-treated cells compared with control at the early stages of hypoxia exposure (within first 12 h); however, cell viability was higher in 3-MA-treated cells than control after hypoxia for 24 h (Appendix A). Conversely, 3-MA pretreatment increased membrane damage in early stages of hypoxia but then this damage was alleviated after hypoxia for 24 h (Appendix A). These results indicate that autophagy plays different roles in hypoxia-induced H9c2 cell injury over time and is beneficial in early stage of hypoxia but detrimental after 24 h of hypoxia (excessive autophagy), in keeping with previous reports [12]. Hypoxia for 24 h was used in subsequent experiments.

Based on the miR-27a-5p expression pattern and cell injury in hypoxia-exposed H9c2 cells and AMI rat myocardium, we hypothesized that miR-27a-5p is involved in mediating this biological process. To test this hypothesis, gain and loss of function analyses were performed. Effective overexpression and downregulation of miR-27a-5p was achieved in H9c2 cells by transfecting cells with a miR-27a-5p mimics or inhibitor, respectively, after exposure to hypoxia for 24 h (Figure 4A). Overexpression of miR-27a-5p significantly mitigated hypoxic injury, including improved cell viability (*p* < 0.01; Figure 4B), alleviated cell membrane damage (*p* < 0.01; Figure 4C), and reduced cell apoptosis (Figure 4D–F). Meanwhile, miR-27a-5p downregulation yielded the opposite effects (Figure 4D–F). These results demonstrate that miR-27a-5p can reduce hypoxia-induced H9c2 cell injury by inhibiting apoptosis. To further assess the impact of miR-27a-5p on autophagy, the autophagic flux and autophagy-related proteins were assessed in cells exposed to hypoxia for 24 h after transfection. As shown in Figure 4G,H, miR-27a-5p overexpression decreased the level of autophagy, shifted LC3-I expression to LC3-II expression (*p* < 0.05), and increased P62 protein expression compared with control (*p* < 0.01). However, miR-27a-5p downregulation resulted in a more severe autophagy (Figure 4G,H). Altogether, these results indicate that miR-27a-5p has a negative effect on hypoxia-induced autophagy and that miR-27a-5p protects against hypoxia-induced cardiomyocyte injury by reducing apoptosis and excessive autophagy.

### 2.4. Atg7 is The Target of miR-27a-5p

To explore the mechanism underlying miR-27a-5p regulation of excessive autophagy and inhibition of apoptosis, we analyzed candidate target genes of miR-27a-5p using TargetScan (release 7.2) [20] and RNAhybrid 2.2 prediction [21]. The prediction results showed that the 3′-UTR region of *Atg7* mRNA contained a target site for miR-27a-5p, and *Atg7* has been linked to autophagy [22]. We tested the expression of *Atg7* and miR-27a-5p in hypoxia-exposed H9c2 cells and in the main visceral tissues of AMI rat, and then performed a correlation analysis. We found a strongly negative correlation between the expression of *Atg7* and miR-27a-5p in hypoxia-exposed H9c2 cells at different timepoints (*r* = −0.807; Figure 5A). Meanwhile, a moderate negative correlation was observed in AMI rat visceral tissue (*r* = −0.569; Figure 5B). Additionally, overexpression of miR-27a-5p in hypoxia-exposed H9c2 cells significantly reduced *Atg7* mRNA and protein expression, while miR-27a-5p downregulation showed an opposite trend (*p* < 0.01; Figure 5C,D). The aforementioned results suggest that miR-27a-5p alleviates hypoxia-induced cardiomyocyte injury by targeting *Atg7*.

We subsequently performed a dual-luciferase reporter assay to confirm the potential relationship between *Atg7* and miR-27a-5p. The sequence alignment of miR-27a-5p showed high similarity, and likewise miR-27a-5p-binding site in *Atg7* 3′-UTR among several representative species were also conserved, which suggested the conservative interaction mechanism of miR-27a-5p-*Atg7* pair among species (Figure 5E). *Atg7* 3′-UTR containing the miR-27a-5p binding site (WT or MUT) was inserted into dual luciferase plasmid (pmirGLO-*Atg7*-3′-UTR) (Figure 5E). HeLa cells were co-transfected with the WT or MUT recombinant plasmid and miR-27a-5p mimics. Luciferase activity was detected 48 h after transfection. As shown in Figure 5F, co-transfection of miR-27a-5p and WT pmirGLO reporter significantly inhibited luciferase activity compared with the negative control (0.462 fold-change, *p* < 0.01). This effect was eliminated with the MUT pmirGLO reporter, which indicates that *Atg7* is a direct target for miR-27a-5p. A standard validation reporting for miR-27a-5p-*Atg7* interaction in this study is shown in Appendix A [23,24].

## 3. Discussion

In recent years, miRNAs have frequently been reported in cardiovascular disease and play important roles in ischemic heart diseases by regulating the process of autophagy and apoptosis [4]. miR-27a-5p belongs to the miRNA-23a-27a-24 cluster that is reported to be involved in many cardiac diseases [25]. miR-24 has been shown to attenuate mouse AMI and reduces cardiac dysfunction by inhibiting cardiomyocyte apoptosis [26]; miR-23a has been shown to positively regulate cardiac hypertrophy by targeting anti-hypertrophic factor *MuRF1* [27] and *Foxo3a* [28]. Although miRNA-27a has been shown to be involved in the regulation of cardiomyocyte apoptosis, during cardioplegia-induced cardiac arrest through IL10-related pathways [29]; whether it regulates cardiomyocyte survival under hypoxic stress caused by ischemic heart diseases such as AMI, remains to be investigated. Based on previous report that the expression of miR-27a-5p decreased in hypoxia-exposed H9c2 cells, we found in the present study that miR-27a-5p expression likewise decreased in AMI rat myocardium (Figure 3F). More deeply, we revealed the miR-27a-5p-*Atg7* interaction in vivo and in vitro, and functionally, miR-27a-5p attenuated hypoxia-induced cardiomyocyte injury by regulating autophagy and apoptosis via *Atg7*, which further confirmed the crucial roles of miRNA-23a-27a-24 cluster in heart diseases.

Autophagy is an evolutionarily conserved and tightly regulated process that maintains cellular homeostasis in response to stresses, such as hypoxia, by degrading abnormal protein and damaged organelles [30,31]. Nevertheless, autophagy is considered a double-edged sword in the context of AMI, i.e., autophagy may have both damaging and protective roles depending on the hypoxic conditions, such as duration or severity [6,19]. In this study, we found that the degree of autophagy in hypoxia-exposed H9c2 cells increased in a time-dependent manner (Appendix A). Inhibition of autophagy (hypoxia + 3-MA pretreatment) decreased cell viability and increased hypoxia-induced membrane damage compared with control (hypoxia) at the early stages of hypoxia exposure (within first 12 h), however these effects were alleviated after hypoxia for 24 h (Appendix A). These results indicate that autophagy plays different roles in hypoxia-induced H9c2 cell injury over time and is beneficial in early stage of hypoxia but detrimental after 24 h of hypoxia (excessive autophagy), in keeping with previous reports [12]. Thus, elucidating and manipulating the development of cardiomyocyte autophagy under hypoxia may be beneficial to the clinical treatment of ischemic heart diseases.

Acting as the only E1-like enzyme, *Atg7* is located in the hub of the LC3 and Atg12 ubiquitin-like systems and is essential for the expansion of autophagosomal membranes [22]. Accumulating evidence suggests that *Atg7* is not only a crucial marker of autophagy, but also participates in the regulation of cell death and survival [32,33], including in cardiac progenitor cells [34]. Previously, we noted that the expression of miR-27a-5p decreased in acute hypoxia-exposed H9c2 cells using a small RNA-seq, as a known hypoxamiR, however, its underlying function in the cardiomyocyte hypoxic response is unclear [13]. In this study, we showed for the first time, to our knowledge, the negative correlation of miR-27a-5p-*Atg7* pair in vivo and in vitro, and that miR-27a-5p alleviated hypoxia-induced cardiomyocyte injury through regulation of excessive autophagy and apoptosis by inhibiting *Atg7* in vitro. This further highlights miRNA regulation in hypoxia-related heart diseases and may have potential implications for the treatment of ischemic cardiomyopathy in the future. However, the function of miR-27a-5p in hypoxia-induced cardiomyocyte injury is mainly focused on the cell-based experiments in vitro. Thus, animal studies on miR-27a-5p knock in/out, such as CRISP-Cas9-mediated gene editing, may better demonstrate the function of miR-27a-5p in hypoxia-induced cardiomyocyte injury after AMI and this should be performed in future research. In addition, although the sequence in miR-27a-5p and *Atg7* 3’-UTR has high similarity among several representative species, the function and strength of miR-27a-5p and its clinical application in human remain to be further elucidated.

## 4. Materials and Methods

### 4.1. Rat AMI Model

Healthy male Sprague Dawley (SD) rats (308 ± 14 g) were bought from Dashuo Laboratory Animal Center (Chengdu, Sichuan, China) and housed in a standard environment (20 ± 2 °C and 58% ± 2% humidity), with free choice feeding for 1 week before experiment. All animal procedures complied with the Ethics Committee of Sichuan Agricultural University rules (Approval Number DKY-B20171903, 15 February 2018). Coronary artery ligation was performed as previously described, to establish the rat AMI model [35]. Arterial BP and ECG were measured throughout the experiment. A clear elevation of the S-T segment of the ECG indicated successful AMI in the rat (*n* = 9). The same procedure was carried out without coronary artery ligation as sham control (*n* = 9). All rats were anesthetized and euthenized 6 h after coronary artery ligation. Several main visceral tissues were collected and immediately immersed in liquid nitrogen before storing at −80 °C for further experimentation.

### 4.2. H9c2 Cell Culture and Hypoxia Treatment

H9c2 cells (an embryonic rat heart-derived cell line) were routinely maintained in Dulbecco’s Modified Eagle Medium (DMEM) (Hyclone, Logan, UT, USA) with 10% fetal bovine serum (FBS) (GIBCO, Grand Island, NY, USA) at 37 °C in a humidified atmosphere containing 5% CO_2_ and 95% air. To establish hypoxia in vitro, cells with 50% confluency received hypoxia treatment for 24 h in a modular incubator chamber with 5% CO_2_, 1% O_2_ and 95% N_2_ (MIC-101, Billups-Rothenberg, Del Mar, CA). Cells in the normoxic group were placed in conventional conditions (5% CO_2_ and 95% air) and served as the control.

### 4.3. H9c2 Cell Transfection

Specific mimics and inhibitor of miR-27a-5p (RIBOBIO, Guangzhou, Guangdong, China) were transfected in cells at 50% confluency to facilitate gain and loss of function. Three groups of cells were designed; a mimic, an inhibitor and a negative control. Transfection solutions were premixed and added to the medium at a final concentration of 50 nM (or 100 nM for the inhibitor) using Lipofectamine 2000 (Invitrogen, Grand Island, NY, USA) in accordance with the manufacturer’s protocol. After 6 h in the transfection medium, all groups were replaced with new medium before receiving hypoxia treatment for 24 h for subsequent experimentation.

### 4.4. Cell Counting Kit-8 (CCK8) and Lactate Dehydrogenase (LDH) Release Assay

To evaluate hypoxia-induced cell injury, cell viability and LDH release were analyzed using a CCK8 and a LDH Cytotoxicity Assay Kit (Beyotime, Shanghai, China), respectively. H9c2 cells were cultured in 96-well plate and received the relevant treatments (such as hypoxia, transfection) at the given time. For CCK8 detection, 10 μL CCK8 reagent was added to the culture medium 4 h before analysis. Optical density (OD)_450_ values were measured using a microplate reader (Thermo Fisher Scientific, Madrid, Spain). For LDH release analysis, the culture medium in each group was premixed with the relevant reagent and incubated in accordance with the manufacturer’s protocol. OD_490_ values were measured and LDH release rate presented as the percentage of the maximum enzymatic activity. At least three independent experiments were repeated three times. All values are presented as mean ± standard deviation (SD).

### 4.5. Cell Apoptosis Analysis

Cell apoptosis was assessed using an Annexin V-FITC and propidium iodide (PI) detection kit (BD Pharmingen, San Diego, CA, USA), in accordance with the manufacturer’s protocols. Briefly, cells were digested by trypsin and gently washed with phosphate buffered saline (PBS). Cells were then incubated with Annexin V and PI for 10 min at room temperature and assessed by flow cytometry (Beckman Coulter, Brea, USA). The raw data were analyzed using CytExpert 2.0 software and more than 10,000 cells in each group were used for statistical analysis. All values are presented as mean ± SD.

### 4.6. HE Staining and Fluorescence Staining of Apoptosis

Tissue sections of the left ventricle were assessed using HE staining. In brief, the rat myocardium was fixed with 4% paraformaldehyde at room temperature, followed by dehydration and embedding in paraffin. The sections were prepared and successively stained using eosin and hematoxylin (Beyotime, Shanghai, China). To observe cell apoptosis, fluorescence staining of H9c2 cells was performed using an apoptosis and necrosis assay kit (Beyotime, Shanghai, China) in accordance with the manufacturer’s instructions. Stained tissue sections and cells were imaged using an Olympus IX53 microscope (Olympus, Tokyo, Japan).

### 4.7. Detection of Autophagosome Formation

H9c2 cells were plated on coverslips. GFP-LC3 plasmids (Beyotime, Shanghai, China) were transfected into H9c2 cells at 50% confluency, before miRNA transfection and exposure to hypoxia. Afterwards, the cells were fixed with 10% formalin and GFP-LC3 fluorescence punctae were imaged using a confocal fluorescence microscope (Olympus, Tokyo, Japan).

### 4.8. Luciferase Reporter Assay

Luciferase activity assays were performed to validate the potential relationship between miR-27a-5p and *Atg7*. Briefly, HeLa cells were routinely maintained in DMEM with 10% FBS at 37 °C. We synthesized the *Atg7* 3′-UTR sequence containing the miR-27a-5p binding site (WT or MUT) and then cloned into the MCS of pmirGLO plasmid (Figure 5E). The WT or MUT recombinant pmirGLO vector was cotransfected with miR-27a-5p mimic or negative control into HeLa cells using Lipofectamine 3000 (Invitrogen, Grand Island, NY, USA), in accordance with the manufacturer’s instructions. Dual luciferase activity was tested by Luciferase Dual Assay Kit (Promega, Madison, WI, USA) 48 h after transfection. Luciferase activity is expressed as an adjusted value (firefly normalized to renilla).

### 4.9. Total RNA Extraction and qRT-PCR

Total RNA was extracted from tissue or cultured cells using HiPure Total RNA Mini Kit (Magen, Guangzhou, China). The quality of total RNA was assessed by NanoDrop 2000 (Thermo Fisher Scientific, Wilmington, DE) and gel electrophoresis. The reverse transcription of mRNA and miRNA from the qualified total RNA was performed using PrimeScript™ RT Reagent Kit (Takara, Beijing, China) and Mir-X™ miRNA First Strand Synthesis Kit (Clontech, Mountain View, USA), respectively, according to the manufacturers’ protocols. qPCR reactions were prepared using an SYBR Premix Ex Taq kit (Takara, Beijing, China) and performed in a Bio-Rad CFX96 PCR System (Bio-Rad, Hercules, USA). The relative expression of mRNA and miRNA was calculated using the 2^−ΔΔCt^ method and expressed as fold-change relative to the corresponding control. *GAPDH* and *U6* served as the reference genes for miRNA and mRNA, respectively. All primers used for qPCR are listed in Appendix A.

### 4.10. Western Blot Analysis

Western blot analysis was performed as previously described [36]. Total protein was extracted from the H9c2 cells and rat myocardium using radioimmunoprecipitation assay lysis buffer containing protease and phosphatase inhibitors (Beyotime, Beijing, China) and quantified using a BCA protein assay. Approximately 30 g of protein was loaded and separated on an 8% SDS-PAGE gel, and then transferred to polyvinylidene difluoride membranes (BIO-RAD, Hercules, USA). The membranes were blocked with nonfat milk for 2 h at room temperature, and then incubated with primary antibodies at 4 °C overnight. Subsequently, the membranes were washed in PBS with Tween-20 before incubating with secondary antibodies for 2 h at room temperature. The antigen–antibody bands were visualized and quantified using ImageJ software (Bethesda, MA, USA). The primary antibodies used in this study and corresponding dilution ratios were as follows: anti-alpha Tubulin (1:1000), anti-Atg7 (1:500), anti-LC3 (1:1000), anti-P62 (1:1000), anti-HIF-1α (1:1000) (Abcam, Cambridge, USA).

### 4.11. Statistical Analysis

All experiments were performed as at least three independent experiments with three technical repetitions. The data are expressed as mean ± SD. Significance tests were performed using SPSS 22.0 software (SPSS, Chicago, USA). Unpaired Student’s *t*-test and one-way ANOVA with Tukey’s post-hoc test were used to evaluate the differences between two groups or three or more groups, respectively. The ∆ BP were used as a surrogate measure of effect to perform a *post hoc* power analysis. The parameters “(*n* = 9, $d=\frac{\left|\mu 1-\mu 2\right|}{\rho }$, sig.level = 0.05, power = , type = “two.sample”, alternative = “two.sided”)” were performed with R (Version 3.2.0) computed by the *pwr* package [37]. *p* < 0.05 was considered as statistically significant (* *p* < 0.05, ** *p* < 0.01).

## 5. Conclusions

We have shown that AMI-induced hypoxia causes cell injury and the expression of miR-27a-5p is decreased in hypoxia-exposed H9c2 cells and AMI rat myocardium. miR-27a-5p attenuates hypoxia-induced cardiomyocyte injury by inhibiting excessive autophagy and apoptosis via *Atg7*. Our findings show that miR-27a-5p has a cardioprotective effect on hypoxia-induced H9c2 injury, and may serve as a novel target for the treatment of hypoxia-related heart diseases.

## Figures and Tables

**Figure 1 ijms-20-02418-f001:**
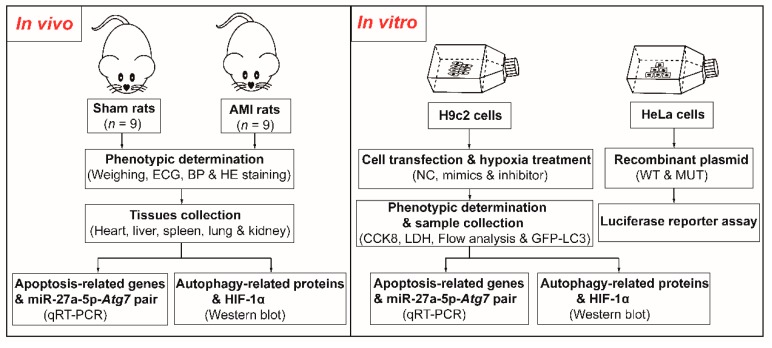
The flow chart of this study. ECG, electrocardiogram; BP, blood pressure; HE staining, hematoxylin & eosin staining; qRT-PCR, quantitative reverse-transcription polymerase chain reaction NC, negative control; CCK8, cell counting kit-8; LDH, lactate dehydrogenase; WT/MUT, wild-type/mutant.

**Figure 2 ijms-20-02418-f002:**
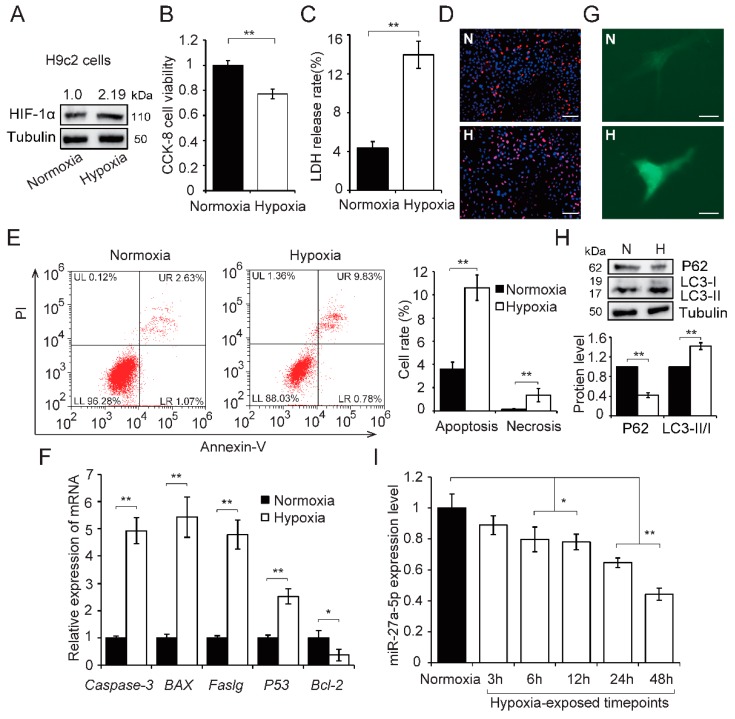
Hypoxia induces H9c2 cell injury and downregulation of miR-27a-5p. H9c2 cells were cultured under hypoxia or normoxia for 24 h. HIF-1α protein increased in H9c2 cells after hypoxia (**A**). Cell viability (**B**), membrane damage (**C**), and cell apoptosis (**D**–**F**) were evaluated by CCK8 assay, LDH release assays, apoptosis staining (scale bar: 50 μm), flow cytometry, and qRT-PCR analysis, respectively. H9c2 cells were transfected with GFP-LC3 plasmids and exposed to hypoxia for 24 h, fluorescence was observed by confocal fluorescence microscopy (**G**); scale bar: 5 μm. The autophagy-related proteins were detected by western blot (**H**). The expression of miR-27a-5p was tested using qRT-PCR at different hypoxia-exposed timepoints (**I**). Three independent experiments were performed in triplicate. Data are expressed as the mean ± SD. * *p* < 0.05, ** *p* < 0.01. N: normoxia; H: hypoxia.

**Figure 3 ijms-20-02418-f003:**
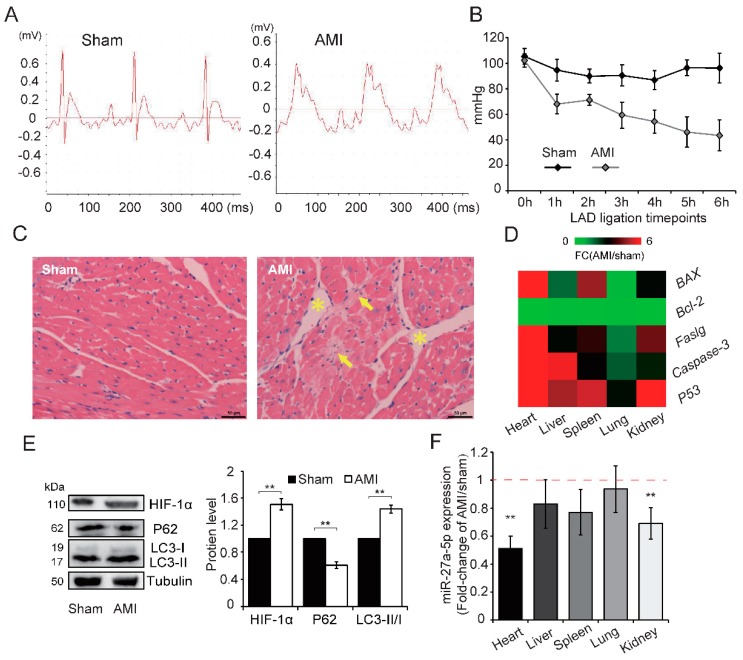
AMI widely induces injury and reduces miR-27a-5p expression in rats. A model of AMI was established in rats by ligating the coronary artery, and confirmed by analyzing ECG (**A**) and BP (**B**). (**C**) HE staining showed morphological differences between sham and AMI rats in coronal sections of the left ventricle; Yellow arrowheads and asterisks highlight local necrosis and intercellular gaps, respectively; scale bar: 50 μm. Expression patterns of apoptosis-related genes (**D**) in main visceral tissues (including heart, liver, spleen, lung and kidney) were determined by qRT-PCR (AMI *vs* Sham). AMI increased HIF-1α expression and promoted the conversion of LC3-I to LC3-II, but decreased P62 expression (**E**). miR-27a-5p expression (**F**) in main visceral tissues by qRT-PCR analysis (AMI *vs* Sham). Data are presented as the means ± SD of three independent experiments. ** *p* < 0.01. LAD: left anterior descending coronary artery.

**Figure 4 ijms-20-02418-f004:**
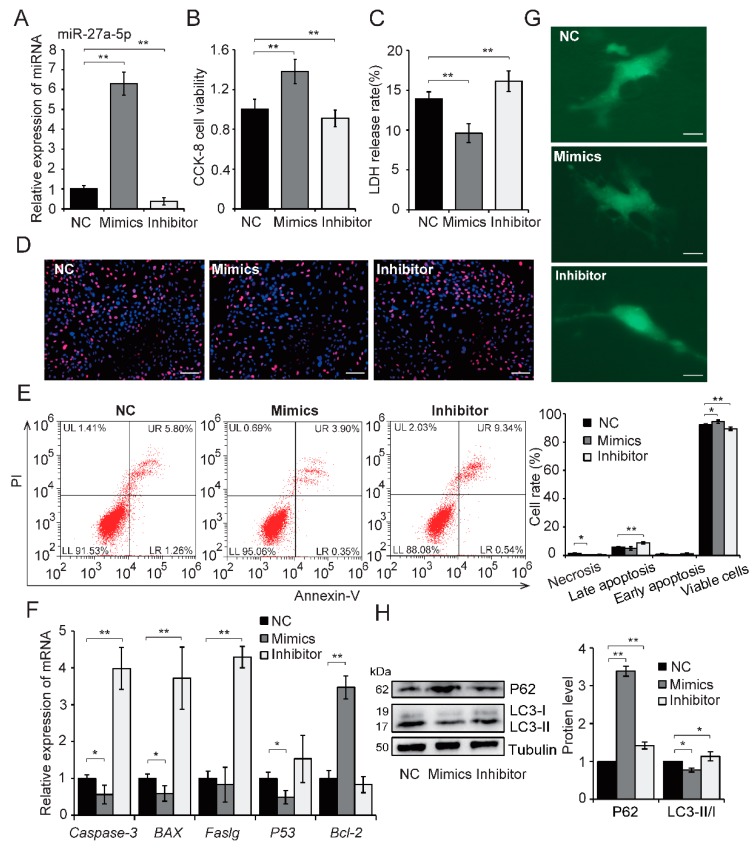
miR-27a-5p attenuates hypoxia-induced excessive autophagy and apoptosis in H9c2 cells. H9c2 cells were exposed to hypoxia for 24 h after transfection of a specific miR-27a -5p mimics or inhibitor. Transfection efficiency was analyzed by qRT-PCR (**A**). Cell viability (**B**), membrane damage (**C**), and cell apoptosis (**D**–**F**) were assessed by CCK8 assays, LDH release assays, apoptosis staining (scale bar: 50 μm), flow cytometry and qRT-PCR analysis, respectively. The level of autophagy was evaluated by GFP-LC3 fluorescence after hypoxia for 24 h (**G**); scale bar: 5 μm. Autophagy-related proteins were detected by western blot (**H**). Three independent experiments were performed in triplicate. Data are expressed as the mean ± SD. * *p* < 0.05, ** *p* < 0.01. NC: negative control.

**Figure 5 ijms-20-02418-f005:**
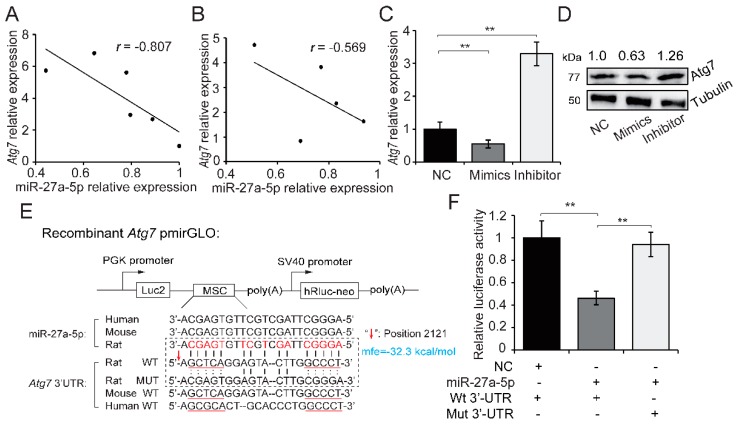
*Atg7* is a direct target of miR-27a-5p. Relative expression correlation analysis between miR-27a-5p and *Atg7* during hypoxia at different timepoints (0, 3, 6, 12, 24 and 48 h after hypoxia) in H9c2 cells (**A**), and in AMI/sham rat visceral tissues (**B**). mRNA (**C**) and protein (**D**) expression of *Atg7* was tested by qRT-PCR and western blotting after miR-27a-5p gain and loss of function in hypoxia-exposed H9c2 cells. (**E**) Schematic diagram showing the structure of dual-luciferase reporter plasmid pmirGLO and the sequence alignment of miR-27a-5p and *Atg7* 3′-UTR among several representative species (human, mouse and rat). *Atg7* 3′-UTR containing the miR-27a-5p binding site (WT or MUT) was inserted into the multiple cloning site (MSC) of pmirGLO plasmid. (**F**) Luciferase activity was analyzed after co-transfection of recombinant plasmid (WT or MUT) with miR-27a-5p mimic or control into HeLa cells. Three independent experiments were performed in triplicate. Data are expressed as the mean ± SD. ** *p* < 0.01. NC: negative control; mfe: minimum free energy.

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
