# Peer review of "miR-27a-5p Attenuates Hypoxia-induced Rat Cardiomyocyte Injury by Inhibiting Atg7"

_ijms, 2019, doi:10.3390/ijms20102418_

Reviewer 1 Report

The authors investigated the function of miR-27a-5p in cardiomyocyte response to hypoxia. The investigations were carried out in

vivo in rats using an infarct model and in vitro cell culture experiments. H9c2 cell culture with hypoxia treatment were used to measure hypoxia-induced cell injury (CCK8 and LDH assay), cell apoptosis (Annexin V-FITC and propidium iodide detection kit, fluorescence staining) and autophagosome formation (GFP-LC3 plasmid transfection). Furthermore HeLa cells and Luciferase activity assay were used to determine the relationship between miR-27a-5p and Atg7. RNA was extracted and measured by qRT-PCR from tissue and from cell culture. Additionally, WesternBlot analyzes were performed. The results show that hypoxia by AMI causes cell injury and decreased expression of miR-27a-5p. Furthermore, miR-27-5p is able to attenuate hypoxia-induced injury by

inhibiting autophagy and apoptosis via Atg7. Thus, the authors concluded that miR-27a-5p has a cardioprotective effect.

Comments to the authors:

1) How can the effect of 3-MA as an autophagy inhibitor be explained? Please add a reference.

2) How long was ischemia performed by ligature in the AMI model?

3) Are there any considerations or results regarding what happens to the cells or the heart in vivo after hypoxia? So, for example, after a reperfusion/reoxygenation?

4) Can the authors please provide a power calculation in the statistical analysis section.

Line 152:  Please correct the sentence: “Autophagy has a multiple effects in AMI,…“

Author Response

Detailed responses to Reviewers

Below, all critique and suggestions provided by reviewers are cited in gray italics, and our responses are in black. All revisions in the

 manuscript are marked in red using the Word Track Changes.

Reviewer 1:

The authors investigated the function of miR-27a-5p in cardiomyocyte response to hypoxia. The investigations were carried out in vivo in rats using an infarct model and in vitro cell culture experiments. H9c2 cell culture with hypoxia treatment were used to measure hypoxia-induced cell injury (CCK8 and LDH assay), cell apoptosis (Annexin V-FITC and propidium iodide detection kit, fluorescence staining) and autophagosome formation (GFP-LC3 plasmid transfection). Furthermore HeLa cells and Luciferase activity assay were used to determine the relationship between miR-27a-5p and Atg7. RNA was extracted and measured by qRT-PCR from tissue and from cell culture. Additionally, WesternBlot analyzes were performed. The results show that hypoxia by AMI causes cell injury and decreased expression of miR-27a-5p. Furthermore, miR-27-5p is able to attenuate hypoxia-induced injury by inhibiting autophagy and apoptosis via Atg7. Thus, the authors concluded that miR-27a-5p has a cardioprotective effect.

Comments to the authors:

Comment 1-1:

1) How can the effect of 3-MA as an autophagy inhibitor be explained? Please add a reference.

Response 1-1:

Thanks for your review. Autophagy is considered a double-edged sword in the context of AMI, and autophagy in early stage of AMI is beneficial to cardiomyocytes survival but excessive autophagy after AMI will induces autophagic cell death (Hongxin et al). 3-MA is a widely-used autophagy inhibitor in vitro (Seglen et al). In this study, we found that the degree of autophagy in hypoxia-exposed H9c2 cell increased in a time-dependent manner (Figure S1B). Inhibition of autophagy (Hypoxia + 3-MA pretreatment) decreased cell viability and increased hypoxia-induced membrane damage compared with control (Hypoxia) at the early stages of hypoxia exposure (within first 12 h), however these effects were alleviated after hypoxia for 24 h (Figure S1D). These results indicate that autophagy plays different roles in hypoxia-induced H9c2 cell injury over time and is beneficial in early stage of hypoxia but detrimental after 24 h of hypoxia (excessive autophagy), in keeping with previous reports (Liu et al). Our research purposes is to investigate the potential function of miR-27a-5p in hypoxia-induced excessive autophagy and apoptosis. Thus the comparison of control (Hypoxia) and inhibition of autophagy (Hypoxia + 3-MA pretreatment) provide an opportunity that revealing the different roles of autophagy in hypoxia-induced H9c2 cell injury over time. The identified timepoint “hypoxia for 24 h” was used in subsequent experiments.

Hongxin, Z.; Paul, T.; Johnstone, J. L.; Yongli, K.; Shelton, J. M.; Richardson, J. A.; Vien, L.; Beth, L.; Rothermel, B. A.; Hill, J. A. Cardiac autophagy is a maladaptive response to hemodynamic stress. J Clin Invest 2007, 117, 1782-1793.

Seglen, P. O., & Gordon, P. B. 3-Methyladenine: specific inhibitor of autophagic/lysosomal protein degradation in isolated rat hepatocytes. PNAS 198279, 1889-1892.

Liu, X.; Deng, Y.; Xu, Y.; Jin, W.; Li, H. MicroRNA-223 protects neonatal rat cardiomyocytes and H9c2 cells from hypoxia-induced apoptosis and excessive autophagy via the Akt/mTOR pathway by targeting PARP-1. J Mol Cell Cardiol 2018, 118, 133-146.

Comment 1-2:

2) How long was ischemia performed by ligature in the AMI model?

Response 1-2:

In this study, coronary artery ligation was performed to establish the rat AMI model, and all rats were anesthetized and sacrificed 6 h after coronary artery ligation (Xin et al). Please see the revised manuscript as below (Page 10, Lines 274-285).

4.1. Rat AMI Model 

Healthy male Sprague Dawley (SD) rats (308 ± 14 g) were bought from Dashuo Laboratory Animal Center (Chengdu, Sichuan, China) and housed in a standard environment (20 ± 2 °C and 58% ± 2% humidity), with free choice feeding for 1 week before experiment. All animal procedures complied with the Ethics Committee of Sichuan Agricultural University rules. Coronary artery ligation was performed as previously described, to establish the rat AMI model [34]. Arterial BP and ECG were measured throughout the experiment. A clear elevation of the S-T segment of the ECG indicated successful AMI in the rat (n = 9). The same procedure was carried out without coronary artery ligation as sham control (n = 9). All rats were anesthetized and sacrificed 6 h after coronary artery ligation. Several main visceral tissues were collected and immediately immersed in liquid nitrogen before storing at −80 °C for further experimentation.

Xin, L.; Baoqiu, W.; Hairong, C.; Yue, D.; Yang, S.; Lei, Y.; Qi, Z.; Fei, S.; Dan, L.; Chaoqian, X. Let-7e replacement yields potent anti-arrhythmic efficacy via targeting beta 1-adrenergic receptor in rat heart. J Cell Mol Med 2014, 18, 1334-1343.

Comment 1-3:

3) Are there any considerations or results regarding what happens to the cells or the heart in vivo after hypoxia? So, for example, after a reperfusion/reoxygenation?

Response 1-3:

Thanks this review. In this study, we established AMI model in rat by ligating coronary artery. A clear elevation of the S-T segment of the ECG and continuous decline of arterial blood pressure indicated successful AMI. After coronary artery ligation for 6 h, HIF-1α expression (Figure 3E) and apoptosis level (Figure 3D & Figure S1A) increased in heart, meanwhile, HE staining of the left ventricle showed that the cells in sham rat hearts were arranged uniformly with a normal gap, but local necrosis (indicated by arrowhead) and intercellular gaps (indicated by asterisk) were observed in AMI rats (Figure 3C). These results indicate that AMI induced hypoxia response and severe damage in the rat myocardium. This study investigated the effects of acute hypoxia on cardiomyocytes after AMI, and did not involve reperfusion. In future study, we will continue to explore specific functions after reperfusion.

Comment 1-4:

4) Can the authors please provide a power calculation in the statistical analysis section.

Line 152:  Please correct the sentence: “Autophagy has a multiple effects in AMI,…“

Response 1-4:

Thanks for your suggestion. We used the ∆ BP as a surrogate measure of effect to perform a post hoc power analysis. The results showed a power of > 0.90 with p = 0.05 in every LAD ligation timepoint. We have added these contents in the revised manuscript as below.

A post hoc power analysis of ∆ BP obtained a power of > 0.90 with p = 0.05 in every LAD ligation timepoint (see “Statistical Analysis” for details on power analysis) (Table S1). (“Results”, Page 4, Lines 111-113; “Supplementary Materials”, Page 2, Line 20)

Table S1. Overview of BP changes   in different ligation timepoints

Ligation timpoints

0 h

1 h

2 h

3 h

4 h

5 h

6 h

Mean (Sham)

105.35

94.55

89.82

90.43

86.75

96.4

96.17

Mean (AMI)

102.22

67.97

71.08

59.32

54.22

46.02

43.39

SD (Sham)

6.05

8.41

5.46

8.20

7.54

6.28

11.75

SD (AMI)

5.22

7.79

4.26

10.14

9.05

11.98

12.15

Cohen's d

0.5539567

3.2790809

3.8269133

3.3737619

3.9054956

5.2674025

4.416118

Power

0.1975046

0.9999997

1

0.9999988

1

1

1

Note: BP, blood pressure; Cohen's d is used to evaluate the effect size.

The ∆ BP were used as a surrogate measure of effect to perform a post hoc power analysis. The parameters “(n = 9, , sig.level = 0.05, power = , type = "two.sample", alternative = "two.sided")” were performed with R (Version 3.2.0) computed by the pwr package [36].

(“Materials and Methods”, Page 12, Lines 376-379)

We have corrected the sentence in the revised manuscript as below (Page 6, Lines 143-145).

Autophagy have bidirectional effects in AMI, as autophagy may have both damaging and protective roles depending on the hypoxic conditions, such as duration or severity [19].

Reviewer 2 Report

The manuscript by Zhang et al.

Title: miR-27a-5p attenuates hypoxia-induced rat cardiomyocyte injury by inhibiting Atg7

Main aim and result: Authors investigated the function of miR-27a-5p in the cardiomyocyte response to hypoxia, and showed that miR-27a-5p expression was downregulated in the H9c2 cells at different hypoxia-exposed timepoints and the myocardium of a rat AMI model. miR-27a-5p upregulation attenuated hypoxia-induced cardiomyocyte injury by regulating autophagy and apoptosis via Atg7. Authors validated experimentally that Atg7 is the target of miR-27a-5p.

Conclusion: miR-27a-5p has a cardioprotective effect on hypoxia-induced H9c2 cell injury, suggesting it may be a novel target for the treatment of hypoxia-related heart diseases.

The study is of importance for scientific community and the manuscript is well written. The title and the Summary are informative. Theoretical background is sufficient to explain the reasons why the study was conducted. Materials and methods are appropriate. Conclusion are supported by the presented data.

I have few minor comments for improvement of the study.

Abstract: regulator, change to regulators

Introduction: MiRNAs (microRNAs), change to MicroRNAs (miRNAs)

Methods are adequately described, however, release number for bioinformatics tolls should be added.

Space missing: duration or severity[24].

Please check figures for typos, for example figure 4: ploy(A).

Authors should discuss if this miRNA-target interaction was also experimentally validated in human or if it is predicted using bioinformatics tools.

It is recommended to follow at least main directions from the guidelines for the miRNA terminology and miRNA target reporting standardization: PMID: 26453491 and PMID: 28388300. Both guidelines could also be cited in the Methods section to increase awareness of the community for reporting standardization and to allow faster transfer of the published data to the databases.

Limitations and strengths of the study should be clearly defined. For example, if this is the first report of this miRNA-target interaction.

Author Response

Detailed responses to Reviewers

Below, all critique and suggestions provided by reviewers are cited in gray italics, and our responses are in black. All revisions in the manuscript are marked in red using the Word Track Changes.

Reviewer 2:

Title: miR-27a-5p attenuates hypoxia-induced rat cardiomyocyte injury by inhibiting Atg7

Main aim and result: Authors investigated the function of miR-27a-5p in the cardiomyocyte response to hypoxia, and showed that miR-27a-5p expression was downregulated in the H9c2 cells at different hypoxia-exposed timepoints and the myocardium of a rat AMI model. miR-27a-5p upregulation attenuated hypoxia-induced cardiomyocyte injury by regulating autophagy and apoptosis via Atg7. Authors validated experimentally that Atg7 is the target of miR-27a-5p.

Conclusion: miR-27a-5p has a cardioprotective effect on hypoxia-induced H9c2 cell injury, suggesting it may be a novel target for the treatment of hypoxia-related heart diseases.

The study is of importance for scientific community and the manuscript is well written. The title and the Summary are informative. Theoretical background is sufficient to explain the reasons why the study was conducted. Materials and methods are appropriate. Conclusion are supported by the presented data.

I have few minor comments for improvement of the study.

Comment 2-1:

Abstract: regulator, change to regulators

Response 2-1:

Thank you for your meticulous review. We have modified it in our revised manuscript as below (Page 1, Line 23).

MiRNAs, which are a class of posttranscriptional regulators, have been shown to be involved in the development of ischemic heart diseases.

Comment 2-2:

Introduction: MiRNAs (microRNAs), change to MicroRNAs (miRNAs)

Response 2-2:

Thank you for your meticulous review. We have modified it in our revised manuscript as below (Page 2, Line 52).

MicroRNAs (miRNAs), a class of highly conserved non-coding RNAs, are major posttranscriptional regulator that involving in almost all cellular processes.

Comment 2-3:

Methods are adequately described, however, release number for bioinformatics tolls should be added.

Response 2-3:

Thank you for your relevant suggestion. The release number for bioinformatics tolls using in this study have added in our revised manuscript as below (Page 7, Lines 186-187).

we analyzed candidate target genes of miR-27a-5p using TargetScan (release 7.2) [20] and RNAhybrid 2.2 prediction [21].

Comment 2-4:

Space missing: duration or severity[24].

Response 2-4:

Thank you for your meticulous review. We have modified it in our revised manuscript as below (Page 6, Line 145).

such as duration or severity [19].

Comment 2-5:

Please check figures for typos, for example figure 4: ploy(A).

Response 2-5:

Thank you for your meticulous review. We have modified “ploy(A)” to “poly(A)” in Figure 5 (Page 8, Line 211), and carefully checked similar typos in the full text again (Note: the number of the figure has been correspondingly changed in revised manuscript).

Comment 2-6:

Authors should discuss if this miRNA-target interaction was also experimentally validated in human or if it is predicted using bioinformatics tools.

Response 2-6:

Thanks for your suggestion. Target prediction of miR-27a-5p using TargetScan (release 7.2) indicated that Atg7 is candidate target gene of miR-27a-5p among several representative species (human, mouse and rat). Sequence alignment of miR-27a-5p showed high similarity, and likewise miR-27a-5p-binding site in Atg7 3’-UTR among these species were also conserved. These suggested the conservative interaction mechanism of miR-27a-5p-Atg7 pair among species. However, miR-27a-5p-Atg7 pair did not reported experimentally in human.

We added this content as independent section in “Discussion” as below (Page 10, Lines 269-272).

In addition, although the sequence high similarity in miR-27a-5p and Atg7 3’-UTR among several representative species, the function and strength of miR-27a-5p and its clinical application in human remain to be further elucidated.

We also supplemented the sequence alignment of miR-27a-5p and Atg7 in “Results” (Page 8, Lines 199-202) and Figure 5E (Page 8, Lines 211-222). As below:

The sequence alignment of miR-27a-5p showed high similarity, and likewise miR-27a-5p-binding site in Atg7 3’-UTR among several representative species were also conserved, which suggested the conservative interaction mechanism of miR-27a-5p-Atg7 pair among species (Figure 5E).  

Figure 5. Atg7 is a direct target of miR-27a-5p. Relative expression correlation analysis between miR-27a-5p and Atg7 during hypoxia at different timepoints (0, 3, 6, 12, 24 and 48 h after hypoxia) in H9c2 cells (A), and in AMI/sham rat visceral tissues (B). mRNA (C) and protein (D) expression of Atg7 was tested by qRT-PCR and western blotting after miR-27a-5p gain and loss of function in hypoxia-exposed H9c2 cells. (E) Schematic diagram showing the structure of dual-luciferase reporter plasmid pmirGLO and the sequence alignment of miR-27a-5p and Atg7 3’-UTR among several representative species (human, mouse and rat). Atg7 3′-UTR containing the miR-27a-5p binding site (WT or MUT) was inserted into the multiple cloning site (MSC) of pmirGLO plasmid. (F) Luciferase activity was analyzed after co-transfection of recombinant plasmid (WT or MUT) with miR-27a-5p mimic or control into HeLa cells. Three independent experiments were performed in triplicate. Data are expressed as the mean ± SD. ** p < 0.01. NC: negative control.

Comment 2-7:

It is recommended to follow at least main directions from the guidelines for the miRNA terminology and miRNA target reporting standardization: PMID: 26453491 and PMID: 28388300. Both guidelines could also be cited in the Methods section to increase awareness of the community for reporting standardization and to allow faster transfer of the published data to the databases.

Response 2-7:

Thanks for your relevant suggestion. According to miRBase 22 release, the -5’ strand of mature miR-27a, also called miR-27a* (Previous IDs), are processed from the pre-mir-27a and usually named miR-27a-5p in many recent studies (Mizuno, et al. & Regis, et al).

Mizuno, K., Mataki, H., Arai, T., et al. (2017). The microRNA expression signature of small cell lung cancer: tumor suppressors of miR-27a-5p and miR-34b-3p and their targeted oncogenes. Journal of human genetics, 62(7), 671.

Regis, S., Caliendo, F., Dondero, A., et al. (2017). TGF-β1 downregulates the expression of CX3CR1 by inducing miR-27a-5p in primary human NK cells. Frontiers in immunology, 8, 868.

Klara, et al. provide a standards for reporting miRNA-target interactions and we organized relevant content and summarized them in Table S3. Please see in the revised manuscript (Page 8, Lines 209-210) and Supplementary Materials (Table S3, Page 2, Lines 23-24). As below:

A standard validation reporting for miR-27a-5p-Atg7 interaction in this study is shown in Table S3 [23, 24].

Table S3. Validation reporting for miR-27a-5p-Atg7 interaction

1

miRNA

Gene name

Mir27a

Entrez ID

100314006

2

Target gene

Gene name

Atg7

Entrez ID

312647

Transcript

Atg7-201

(if available)

(ENSRNOT00000067532.2)

3

Species

Species

Rattus norvegicus

Species ID

10116

4

Experimental   validation of miRNA–target interaction

Sequence of the   target region, 5'-3'

GCCCT

Genomic location of   MTI / 3'UTR

4:146776236-146776240   /

3'UTR: 25-29

Method for   experimental validation

luciferase reporter   assay,

qPCR, western blot

Tissue, cell lines

heart; H9c2 cell   lines

5

Sequence variant

rs number (synonym)

na

6

Associated disease   or phenotype

As named in the   reference

Acute myocardial   ischemia;

hypoxia treatment

DOID (if available)

na

7

Reference

Author, year

na

PMID

na

Note: na, not available.

Comment 2-8:

Limitations and strengths of the study should be clearly defined. For example, if this is the first report of this miRNA-target interaction.

Response 2-8:

Thanks for your relevant suggestion. We added the limitations and strengths of this study as independent section “Discussion” in revised manuscript as below (Page 9-10, Lines 260-272).

In this study, we showed for the first time, to our knowledge, the negative correlation of miR-27a-5p-Atg7 pair in vivo and in vitro, and that miR-27a-5p alleviated hypoxia-induced cardiomyocyte injury through regulation of excessive autophagy and apoptosis by inhibiting Atg7 in vitro. This further highlights miRNA regulation in hypoxia-related heart diseases and may have potential implications for the treatment of ischemic cardiomyopathy in the future. However, the function of miR-27a-5p in hypoxia-induced cardiomyocyte injury are mainly focused on the cell-based experiments in vitro. Thus, animal studies on miR-27a-5p knock in/out, such as CRISP-Cas9-mediated gene editing, may better demonstrate the function of miR-27a-5p in hypoxia-induced cardiomyocyte injury after AMI and this should be performed in future research. In addition, although the sequence high similarity in miR-27a-5p and Atg7 3’-UTR among several representative species, the function and strength of miR-27a-5p and its clinical application in human remain to be further elucidated.

Reviewer 3 Report

The main aim of the present experimental model was to investigate the potential function of miR-27a-5p in the cardiomyocyte response to hypoxia.

Results extend previous reports from the same Authors.

The topic is intriguing and the manuscript is well written.

I have the following suggestions to improve the overall quality of the manuscript:

1) the Authors used an AMI group; it was compared with sham; furthermore, the Authors evaluated the effects of hypoxic conditions; a figure depicting the flow-chart of interventions through the study is encouraged.

2) Please, avoid to merge results with discussion (page 3). Manuscript should be re-arranged.

3) Please, expand the section on the clinical impact of the results.

Author Response

Detailed responses to Reviewers

Below, all critique and suggestions provided by reviewers are cited in gray italics, and our responses are in black. All revisions in the manuscript are marked in red using the Word Track Changes.

Reviewer 3:

The main aim of the present experimental model was to investigate the potential function of miR-27a-5p in the cardiomyocyte response to hypoxia.

Results extend previous reports from the same Authors.

The topic is intriguing and the manuscript is well written.

I have the following suggestions to improve the overall quality of the manuscript:

Comment 3-1:

1) the Authors used an AMI group; it was compared with sham; furthermore, the Authors evaluated the effects of hypoxic conditions; a figure depicting the flow-chart of interventions through the study is encouraged.

Response 3-1:

Thank you for your relevant suggestion. We have added a figure that depicting the flow chart of our study in the revised manuscript as below. (Page 2, Lines 73) & (Page 3, Lines 78-79)

In this study, we established a model of hypoxia in H9c2 cells and developed an AMI model in the rat to investigate the miR-27a-5p expression pattern in H9c2 cells and the main visceral tissues of rats (Figure 1).

Figure 1. The flow chart of this study.

Comment 3-2:

2) Please, avoid to merge results with discussion (page 3). Manuscript should be re-arranged.

Response 3-2:

We have re-arranged the manuscript and made the results and discussion as separate parts. Please see the revised manuscript.

2. Results” (Page 3-9, Lines 80-222) and “3. Discussion” (Page 9-10, Lines 223-272).

Comment 3-3:

3) Please, expand the section on the clinical impact of the results.

Response 3-3:

We have added the clinical impact of this study and interspersed in the “3. Discussion” in the revised manuscript. As below:

More deeply, we revealed the miR-27a-5p-Atg7 interaction in vivo and in vitro, and functionally, miR-27a-5p attenuated hypoxia-induced cardiomyocyte injury by regulating autophagy and apoptosis via Atg7, which further confirmed the crucial roles of miRNA-23a-27a-24 cluster in heart diseases. (Page 9, Lines 236-239)

These results indicate that autophagy plays different roles in hypoxia-induced H9c2 cell injury over time and is beneficial in early stage of hypoxia but detrimental after 24 h of hypoxia (excessive autophagy), in keeping with previous reports [12]. Thus, elucidating and manipulating the development of cardiomyocyte autophagy under hypoxia may be beneficial to the clinical treatment of ischemic heart diseases. (Page 9, Lines 249-253)

In this study, we showed for the first time, to our knowledge, the negative correlation of miR-27a-5p-Atg7 pair in vivo and in vitro, and that miR-27a-5p alleviated hypoxia-induced cardiomyocyte injury through regulation of excessive autophagy and apoptosis by inhibiting Atg7 in vitro. This further highlights miRNA regulation in hypoxia-related heart diseases and may have potential implications for the treatment of ischemic cardiomyopathy in the future. However, the function of miR-27a-5p in hypoxia-induced cardiomyocyte injury are mainly focused on the cell-based experiments in vitro. Thus, animal studies on miR-27a-5p knock in/out, such as CRISP-Cas9-mediated gene editing, may better demonstrate the function of miR-27a-5p in hypoxia-induced cardiomyocyte injury after AMI and this should be performed in future research. In addition, although the sequence high similarity in miR-27a-5p and Atg7 3’-UTR among several representative species, the function and strength of miR-27a-5p and its clinical application in human remain to be further elucidated. (Page 9-10, Lines 260-272)

Round  2

Reviewer 3 Report

I have no other comments